# Green Tea Epigallocatechin 3-Gallate Reduced Platelet Aggregation and Improved Anticoagulant Proteins in Patients with Transfusion-Dependent β-Thalassemia: A Randomized Placebo-Controlled Clinical Trial

**DOI:** 10.3390/foods13233864

**Published:** 2024-11-29

**Authors:** Touchwin Petiwathayakorn, Sasinee Hantrakool, Kornvipa Settakorn, Nuntouchaporn Hutachok, Adisak Tantiworawit, Nopphadol Chalortham, Pimpisid Koonyosying, Somdet Srichairatanakool

**Affiliations:** 1Department of Biochemistry, Faculty of Medicine, Chiang Mai University, Chiang Mai 50200, Thailand; touchwinchalae@gmail.com (T.P.); stkornvipa@gmail.com (K.S.); the.pleum@hotmail.com (N.H.); pimpisid.k@cmu.ac.th (P.K.); 2Division of Hematology, Department of Internal Medicine, Faculty of Medicine, Chiang Mai University, Chiang Mai 50200, Thailand; sasinee.h@cmu.ac.th (S.H.); adisak.tan@cmu.ac.th (A.T.); 3Department of Pharmaceutical Sciences, Faculty of Pharmacy, Chiang Mai University, Chiang Mai 50200, Thailand; nopphadol.chalortham@cmu.ac.th

**Keywords:** anti-coagulation, EGCG, green tea, platelet aggregation, thalassemia, thrombosis

## Abstract

Patients with transfusion-dependent β-thalassemia (TDT) with iron overload have been linked to hypercoagulability and increased platelet (PLT) activation that causes thrombosis. Green tea extract (GTE) rich in epigallocatechin-3-gallate (EGCG) exerts iron-chelating and antithrombotic properties. The study aimed to assess the effects of GTE treatment on plasma coagulation state and PLT function in vitro and in patients with TDT. The subjects consumed a placebo or GTE tablets (50 mg and 2 × 50 mg EGCG equivalent) every day for two months. Blood was then collected from the treated patients for analyses of PLT numbers, agonist-induced PLT aggregation, and anti-coagulation proteins. In our findings indicate that the in vitro treatment of GTE (at least 1 mg EGCG equivalent) inhibited PLT aggregation in patients who were healthy and with thalassemia platelet-rich plasma (PRP), which was significant in the healthy PRP. Consistently, GTE treatment inhibited the PLT aggregation that had been ex vivo generated by collagen or ADP. In addition, consumption of GTE tablets greatly inhibited PLT aggregation and increased the plasma levels of proteins C and S, as well as the free protein S concentrations depending upon the time course, but not the GTE dosage. Moreover, plasma ferritin levels decreased in both green tea tablet groups in a time-dependent manner (*p* < 0.05 in the second month). In conclusion, EGCG-rich GTE diminished PLT aggregation in patients who were healthy and patients with thalassemia plasma. It also improved PLT aggregation and hypercoagulability in patients with TDT by increasing the antithrombotic activity of protein C and protein S. This would suggest an adjuvant of GTE could reduce the risk of thrombosis associated with iron overload.

## 1. Introduction

A hypercoagulable condition has been linked to patients with β-thalassemia who experience iron overload, and it is recognized as a major contributor to thrombotic events or thromboembolism [1]. Pathophysiologically, increased platelet (PLT) activation and aggregation, along with the aberrant red blood cell (RBC) membrane surface, may contribute to this condition. This can lead to phosphatidylserine exposure, activated endothelial cells, PLT microparticles, decreases in nitric oxide (NO^•^) levels, and diminished splenectomy and abnormal blood-clotting mechanisms [1]. Therefore, patients with splenectomized β-thalassemia have been suggested to take antithrombosis or antiplatelet medications such as warfarin or aspirin [2], iron chelators such as deferoxamine (DFO), deferiprone (DFP), or deferasirox (DFX) [3], along with antioxidants, such as ascorbic acid, α-tocopherol, or N-acetylcysteine (NAC) [4]. While combination therapy may have negative consequences, it is hoped that the chemicals can improve oxidative stress, iron overload, and thrombosis [5].

Vasculopathy, hemostasis, and thrombosis are linked to PLT hyperactivity and activation. The G protein-coupled purinergic (P) 2X and 2Y receptors are ligand-gated cation channels in which adenosine diphosphate (ADP) is recognized as an agonist for its important function in PLT activation [6]. Thus, antiplatelet drugs are considered mainstay treatments due to some of the pharmacological effects they generate. These would include blocking the purinergic P2Y12 receptor and the thromboxane receptor. However, they can also display some limitations in their action and have been associated with bleeding in patients [7]. The search for possible antiplatelet drugs derived from plant extracts has gained momentum in the quest for complementary therapies. We have recently shown that arabica coffee extracts rich in chlorogenic acid (CGA) have anti-cyclooxygenase (COX) and reactive oxygen species (ROS)-scavenging properties while also inhibiting the concentration-dependent aggregation of PLT in platelet-rich plasma (PRP) that is triggered by ADP, collagen (Col), epinephrine, and arachidonic acid [8].

In addition, green tea extract (GTE) consists mainly of polyphenolics including catechin (C), epicatechin-3-gallate (ECG), epicatechin (EC), epigallocatechin (EGC), epigallocatechin-3-gallate (EGCG), CGA, caffeic acid, chlorogenic acid (CGA) and caffeine (CF), of which EGCG is the most abundant accounting for 59% of total catechins [9]. Interestingly, GTE exerts a range of antioxidant, reactive oxygen species (ROS)-scavenging, and metal-chelating properties [10]. In addition, it is recognized for its ability to inhibit the functions of inducible nitric oxide synthase (iNOS), cyclooxygenase (COX), and lipoxygenase (LOX) [11,12]. Similarly, ex vivo studies involving green tea catechins and EGCG have demonstrated their capability to prevent the aggregation of platelets (PLT) induced by ADP, collagen (Col), thrombin, and thapsigargin in vitro [13], as well as in cases of pulmonary thrombosis in mice [13]. Likewise, it has been reported that EGCG can reverse ADP-induced PLT aggregation and thrombosis by inhibiting the action of platelet phospholipase A_2_ in diabetic rats induced by streptozotocin, indicating its anticoagulant and antiplatelet effects [14,15]. Moreover, ex vivo applications of GTE, which is high in EGCG and tannins, demonstrated a concentration-dependent inhibition of PLT aggregation in rats induced by Col, comparable to certain non-steroidal anti-inflammatory drugs like aspirin (ASA) [7]. In a consistent manner, the ex vivo treatment with EGCG showed a concentration-dependent reduction in the aggregation of human PLT induced by ADP and Col, although it did not affect PLT aggregation induced by ASA, arachidonic acid, clopidogrel, or ticagrelor [15]. The study aimed to assess the effect of EGCG-rich GTE treatment on the coagulation state and PLT aggregation in patients with transfusion-dependent β-thalassemia (TDT).

## 2. Materials and Methods

### 2.1. Chemicals and Reagents

All analytical or the purest grade chemicals and authentic standards were used. Collagen and ADP that were used as agonists were purchased from Helena Laboratories, Beaumont, TX, USA. Every organic solvent was either HPLC grade or the purest possible grade.

### 2.2. Preparation and Chemical Analyses of GTE, Placebo, and GTE Tablets

Herein, GTE was derived from freshly harvested tea shoots (*Camellia sinensis*) using the technique that was previously developed by Settakorn and coworkers [16] and described in Appendix A. Placebo (PB) and GTE tablets were produced as described in Appendix A. Contents of catechin derivatives, CF, ascorbic acid (AA), tocopherols, and tocotrienols in the GTE, GTE tablets, and placebo tablets were determined [17,18,19] and described in Appendix A, respectively.

### 2.3. Ethical Consideration

Ethical approval required for collecting blood from healthy (N) and TDT subjects to investigate the anti-platelet effect of GTE treatment was granted exemption by Emeritus Professor Panja Kulapongs, MD., Chairman of the Ethics Committee of the Faculty of Medicine, Chiang Mai University, Chiang Mai, Thailand (Reference ID: 7575/Study/Code: BIO-2563-07575; Date of approval: 28 September 2020), as is shown in Appendix A.

In addition, the interventional study was conducted in accordance with the Declaration of Helsinki of 1975, which was revised in 2013 (https://www.wma.net/what-we-do/medical-ethics/declaration-ofhelsinki/, accessed on 1 February 2020), and the protocol was approved by the Ethical Committee for Human Study of the Faculty of Medicine, Chiang Mai University (Reference Number: MED-2561-05846; Date of approval: 22 May 2020), as is shown in Appendix A. Informed consent for participation was obtained from all subjects involved in the study (Appendix A).

### 2.4. Clinical Trial Registration

The clinical study protocol (Appendix A) was evaluated and authorized by the committee of the Medical Research Foundation of Thailand at the Thai Clinical Trials Registry (ID: TCTR20211118002; Date of Registration: 18 November 2021) (https://www.thaiclinicaltrials.org/), which is recognized by the World Health Organization International Clinical Trials Registry Platform. The research was carried out in accordance with the reporting standards outlined in the guidelines set by the Consolidated Standards of Reporting Trials (CONSORT) 2010 [20].

### 2.5. In Vitro Study for GTE Treatment on PLT Aggregation

#### 2.5.1. Blood Collection and Preparation

In total, sixteen healthy (N) subjects (seven male and nine female) and twenty patients with thalassemia classified as four with NTDT (one male and three female) and sixteen with TDT (eight male and eight female) were enrolled in this study (Appendix A). Importantly, prior to undergoing subsequent blood transfusions, the patients were asked to stop taking iron chelators and anti-coagulant drugs for 72 h. Fifteen milliliters of venous blood were withdrawn from each subject and transferred into 3.2% sodium citrate-coated vacutainers (BD Vacutainer, Becton-Dickinson Company, Franklin Lakes, NJ, USA). Accordingly, citrate anticoagulant blood samples collected from subjects who were healthy and subjects with thalassemia underwent two separate centrifugation periods: the first at 1000 rpm for 10 min to extract platelet-rich plasma (N-PRP and T-PRP, respectively) for evaluating platelet aggregation, and the second at 3000 rpm for 10 min to obtain platelet-poor plasma (N-PPP and T-PPP, respectively). This process was carried out to evaluate relevant coagulation outcomes, as will be detailed below.

#### 2.5.2. Treatment of PRP with GTE

Initially, 400 µL of N-PRP and T-PRP samples were treated with 50 µL of PBS; 3.125, 6.25, and 12.5 mg/mL of GTE solution (equivalent to 1.25, 2.5, and 5.0 mg EGCG, respectively); or standard EGCG (1.25, 2.5, and 5.0 mg) solution. The treated PRP were then challenged with agonists, such as 5 µM of ADP or 2 µg/mL of Col, and PLT aggregation was measured, as will be described below.

#### 2.5.3. Assay of PLT Aggregation Induced by Agonists

Initially, 450 µL of PRP (test) and PPP (blank) were separately mixed with 50 µL of agonist solution in separate PLT aggregation test cuvettes using a siliconized stir bar. Subsequently, PLT aggregation values were promptly assessed with a platelet aggregometer (Chrono-Log Model 700, Havertown, PA, USA) and then maintained at a temperature of 37 °C for 7 min [8]. As a result, the percentage values were calculated from the aggregation graphs produced by the aggregometer using the compatible Aggrolink8 software for Windows.

### 2.6. Assessment for Consumption of GTE on PLT Aggregation and Blood Coagulation in TDT

#### 2.6.1. Patient Selection

Patients with β-thalassemia who regularly underwent their physical and blood examinations at the Adult Thalassemia Clinic, Maharaj Nakorn Chiang Mai Hospital, Faculty of Medicine, Chiang Mai University, Chiang Mai, Thailand, were enrolled in this study. As for the inclusion criteria, all participants were adult Thai patients with TDT aged between 20 and 65 years who could communicate in Thai, had attended the clinic for regular treatments, and had not received any additional treatments other than blood transfusions (Tx) and iron chelation therapy of Desferal (deferoxamine mesylate, Novartis AG, Basel, Switzerland), Exjade (DFX, Novartis AG, Basel, Switzerland), GPO-L1 (a generic name of DFP manufactured by Government Pharmaceutical Organization, Bangkok, Thailand), or a combination of any of them for a minimum of three months prior to and throughout the study duration. The exclusion criteria included the following: the existence of other major blood disorders, a confirmed history of serious liver or kidney problems, having current infections or fever at the time of enrollment, being pregnant or nursing, an inability to give informed consent, and any involvement in smoking or alcohol use during the study period [16].

#### 2.6.2. Randomization and Study Design

Using the stratified random sampling technique, thirty-three TDT subjects enrolled in the study, but three of them dropped out from the study. The calculation of the sample size was conducted using G*Power software version 3.1.9.7, allowing for comparison between three groups. In this case, a repeated measures ANOVA F test was employed, incorporating within-between interaction mode. The anticipated effect size for the new compound is 0.50, which is considered large. The statistical power is set at 95% with a significance level (α) of 0.05. Measurements were taken at three points: baseline (T0), during the first intervention (T1), and following the second intervention (T2). Consequently, the total sample size determined by G*Power is 54, with 18 participants allocated to each group. A total of thirty patients with TDT were categorized into three groups: group 1 received PB (n = 11), group 2 was administered GTE tablets (50 mg EGCG equivalent) (n = 9), and group 3 received 100 mg of EGCG equivalent, or double the amount of GTE tablets (n = 10). Before the commencement of the trial, the results of genotyping and physical assessments were recorded. They documented participants’ age, height, body weight (BW), body mass index (BMI), and the palpability of the liver and spleen. For 60 days, all participants consumed the product daily and were advised to avoid meals rich in polyphenolic components. Significantly, blood samples were taken on days 0, 30, and 60 after a 72 h pause from their iron chelation therapy and just before their subsequent blood transfusion [16].

#### 2.6.3. Intervention

This study was carried out between 1st July 2020 and 31st May 2021 at the Adult Thalassemia Clinic, and a flow chart illustrating the study was created (Figure 1). It was required of all participants that they cease taking iron chelators for a full 72 h prior to arriving at the thalassemia clinic and undergoing the blood collection process. Accordingly, the PB group, along with the 50 mg EGCG equivalent-GTE group and the group receiving double the 50 mg EGCG equivalent-GTE group, were orally given a designated treatment once daily for a duration of two months. In all instances, the oral administration took place 30 min before breakfast.

#### 2.6.4. Patients’ Blood Collection

At the start of the study, as well as one month and two months after the products were administered, but before subsequent blood transfusions were administered, venous blood (15 mL) was drawn into three different types of vacutainer tubes. These included a vacutainer tube coated with ethylenediamine tetra-acetic acid (EDTA) (BD Vacutainer, Becton-Dickinson Company, Franklin Lakes, NJ, USA) (2 mL), a vacutainer tube coated with 3.2% sodium citrate (3 mL), and a vacutainer tube coated with lithium heparin. All these blood samples underwent further analyses. The EDTA anticoagulant blood was analyzed for a complete blood count (CBC), as is outlined below. As has been described above, the citrate anticoagulant blood tube was subjected to centrifugation to obtain PRP for the determination of PLT aggregation. This was also performed to obtain PPP for the determination of coagulation, which will be described below. The heparin anticoagulant blood tube was also centrifuged at 3000 rpm for a duration of 10 min, and the plasma was separated for the analysis of relevant biochemical parameters, as will be outlined below.

#### 2.6.5. Determination of PLT Numbers

The EDTA anticoagulant blood was assessed for complete blood count (CBC) using an automated hematology analyzer and suitable reagents (Model DxH900, Beckman Coulter, Brea, CA, USA) according to the manufacturer’s instructions. Accordingly, PLT indices were then recorded for further consideration.

#### 2.6.6. Measurements of Blood Coagulation

Based on the guidelines provided by the manufacturer of an automated coagulated analyzer (Beckman ACL 1000, Beckman Coulter, Brea, CA, USA), and using the correct reagents, blood coagulation metrics such as prothrombin time (PT), activated partial thromboplastin time (aPTT), plasma free protein S, as well as protein S and protein C activities, were assessed in PPP samples.

#### 2.6.7. Determinations of Iron Status and Liver Functions

Based on the guidelines provided by the manufacturer, the following parameters and actions were assessed using an automated ClinChem Analyzer (Cobas 8000 modular Series, Roche Diagnostics International AG, Rotkreuz, Switzerland) along with the suitable reagents. These assessments encompassed plasma iron (PI), transferrin saturation (TS), and total iron-binding capacity (TIBC), as well as concentrations of ferritin, total protein, albumin, and globulin, in addition to the activities of aspartate aminotransferase (AST), alanine aminotransferase (ALT), and alkaline phosphatase (ALP) in the plasma [21].

### 2.7. Outcomes of the Study

Primary outcomes of this study involved the assessment of PLT aggregation, blood coagulation time, and plasma coagulation proteins that were determined based on PLT indices and aggregation, PT and aPTT, and plasma levels of protein S and protein C, respectively. A secondary outcome included the measurements of iron status parameters and liver function indicators to evaluate the relationship between the GTE treatment and blood coagulation, as well as PLT aggregation in conjunction with plasma ferritin, SI, % TS values.

### 2.8. Statistical Analysis

The analysis of results was conducted using IBM SPSS Statistics version 21.0 (IBM, Armonk, NY, USA). Data and experimental results have been presented as values of mean ± standard deviations (SD) or mean ± standard errors of the mean (SEM). The demographic information was subjected to descriptive analysis. The distribution’s normality was evaluated using the Kolmogorov–Smirnov test. Group comparisons were conducted at various time points during the study, with results evaluated using repeated measures of one-way analysis of variance (ANOVA) and Bonferroni tests. Notably, *p* < 0.05 was deemed to indicate a significant difference. Before making comparisons, it was assumed that the mean values of each parameter collected from three distinct groups at the study’s onset were not significantly different when employing the ANOVA test. Thus, any missing data were left unaddressed. We have complied with a complete Consolidated Standards of Reporting Trials (CONSORT) 2010 checklist and flow diagram of the randomized trial (Figure 1).

## 3. Results

### 3.1. Chemical Compositions in GTE, GTE, and Placebo Tablets

High-performance liquid chromatography/diode array detection (HPLC/DAD) analysis has demonstrated that standard C, EC, EGCG, and ECG were eluted at the retention times (T_R_) of 1.982, 3.807, 5.205, 6.154, and 13.375 min, respectively (Appendix A). According to our findings, GTE granules showed C, EC, EGCG, and ECG peaks that were eluted equivalently to the authentic standards (Appendix A). In addition, GTE showed a peak of CF eluted at 10.476 min, equivalent to that of standard CF (Appendix A). However, GTE did not reveal any peak of AA when compared with the standard AA that had been eluted at 2.289 min (Appendix A). Moreover, GTE elucidated a small peak of α-tocopherol at 7.360 min, whereas the peaks of standard α-tocopherol, α-tocotrienol, β-tocopherol, γ-tocopherol, β-tocotrienol, γ-tocotrienol, δ-tocopherol, and δ-tocotrienol levels were detected at T_R_ values of 7.202, 8.203, 8.617, 9.742, 10.396, 12.181, 12.756, and 14.796 min, respectively (Appendix A). Accordingly, all compositions present in the GTE granules, GTE, and a placebo tablet have been summarized in Appendix A.

### 3.2. Effect of GTE Treatment on PLT Aggregation In Vitro

#### 3.2.1. Information About the Blood Donors

As has been shown in Table 1, six N subjects (three males and three females within an age range of 25–28 years) and ten patients with TDT (five males and five females within an age range of 20–46 years), who had been diagnosed as three β-thalassemia HbE (BE) and seven patients with β-thalassemia major (BM), were recruited for this study. All the patients with thalassemia received iron chelators according to the hematologists’ prescriptions, of which 70% were splenectomized and 70% were given acetylsalicylate (ASA). Remarkably, PLT numbers and serum ferritin (Ft) levels were higher in patients with TDT than the N subjects. Nonetheless, there were no significant differences in gender, age, BW, and BMI among the two groups.

#### 3.2.2. PLT Aggregation in N-PRP and T-PRP Treated with GTE

Obviously, GTE treatments (1.25–5 mg EGCG equivalent) inhibited PLT aggregation in 5 μM of ADP-induced N-PRP samples (n = 6) significantly independent of dose and gender (3 male and 3 female) when compared with those that had not undergone GTE treatment, while EGCG treatments (1.25–5 mg) were less effective than GTE and more dominant in the male samples than the female samples (Figure 2A). In addition, the GTE treatments significantly inhibited PLT aggregation in 2 µg/mL of Col-induced N-PRP samples (n = 6) independent of the doses. In contrast, the EGCG treatments inhibited PLT aggregation in a dose-dependent manner in which the inhibitions were significantly greater in males than females (Figure 2B).

In contrast, the GTE and EGCG treatments did not influence PLT aggregation in T-PRP (n = 10) induced by 5 μM of ADP (Figure 3A), and the two treatments tended to inhibit PLT aggregation in the T-PRP induced by 2 μg/mL of Col (Figure 3B). When compared with those that had not undergone treatment, the GTE was more effective than the EGCG, whereas the inhibitions were not differentiated in comparisons made between the male T-PRP (n = 5) and the female T-PRP (n = 5) subjects.

#### 3.2.3. Effect of Splenectomy on PLT Aggregation in PRP Treated with GTE

As has been shown in Figure 4A, GTE (1.25–5.0 mg EGCG equivalent) treatments inhibited PLT aggregation significantly in N-PRP, slightly in T-PRP, and hardly in the PRP of patients with splenectomized thalassemia (T-PRP^) who had been induced with 5 μM of ADP. In comparison, the GTE treatments inhibited PLT aggregation significantly in N-PRP induced by 2 μg/mL of Col, while the treatments slightly inhibited PLT aggregation in the T-PRP^ samples but did not influence PLT aggregation in the T-PRP samples (Figure 4B).

#### 3.2.4. Anti-PLT Activity by GTE in PRP Acquired from Different Thalassemia Types

Herein, GTE (1.25–5 mg EGCG equivalent) treatments tended to inhibit PLT aggregation in T-PRP samples of BE1-BE3, but not BM1-BM7 that were induced by 5 μM of ADP agonist (Figure 5A). Interestingly, four of the seven T-PRP samples (BM1, BM2, BM3, and BM6) responded to inhibitions of 2 μg/mL of Col-induced PLT aggregation by the GTE treatments in a concentration-dependent manner (Figure 5B).

### 3.3. Effects of GTE Consumption on PLT Aggregation and Blood Coagulation in Patients with TDT

#### 3.3.1. Demographic Characteristics

Before the trial started, demographic information was collected from the thirty patients with TDT who were recruited (Table 2). The participants included patients with β-thalassemia (43.3%), HbE/β-thalassemia (50.0%), and AE Bart’s disease (6.7%), and the average ages of individuals in each category showed no significant differences. The subjects were under the chelation treatments with DFP alone (80%), DFX alone (3.33%), a combination of DFP + DFO (6.67%), and DFO + DFX (10.0%). Additionally, a low-dose ASA regimen was provided to a portion of the participants who had undergone splenectomy (70.0%) to mitigate the risk of thrombosis. The average age, liver span, body weight, and body mass index were determined to be statistically insignificant and fell within the healthy range for all subjects. Moreover, the initial measurements of hematological parameters, which included platelet count, protein C and protein S activity, free protein S concentration, and iron-related metrics such as plasma iron, total iron-binding capacity, transferrin saturation, and plasma ferritin, showed no significant differences among all patients.

#### 3.3.2. Platelet Indices

In Table 3, the placebo group (PB1–PB11), along with the groups receiving the 50 mg EGCG equivalent-GTE tablet (GTE1–GTE9) and the 100 mg EGCG equivalent-GTE tablet (GTE10–GTE19), exhibited no significant alterations in the levels of platelet numbers, mean platelet volume (MPV), platelet distribution width (PDW), plateletcrit (Pct), and immature platelet fractions (IPF) after consumption of either GTE tablets or a placebo for a period of one to two months.

#### 3.3.3. Coagulation Time and Anticoagulant Proteins

In this study, PT and aPTT values were used to assess coagulation status. The results indicated that there were no significant differences in PT and aPTT values between the placebo group and the two groups receiving GTE tablets (50 and 100 mg EGCG equivalent) at the same time points during the intervention. Additionally, there were no changes in these values following one and two months of product consumption (Figure 6A,B).

In contrast, as illustrated in Figure 6C–E, the activities of protein C and protein S, along with the concentrations of free protein S, remained unchanged in the plasma of TDT subjects who had taken the placebo for two months. Meanwhile, the levels of protein C and protein S activities—three parameters—increased in a time-dependent manner in both GTE tablet groups (50 and 100 mg EGCG equivalents). Thus, the increases in protein C activity in the low-dose GTE tablet group were not statistically significant (*p* = 0.055), while the increases in the high-dose GTE tablet group were notably significant (*p* = 0.0036) when compared to the baseline values at T_0_ (Figure 6C). Similarly, the rise in protein S activity was most pronounced in the second month for both GTE tablet groups (*p* = 0.022 and 0.001, respectively) compared to the baseline values at T_0_ (Figure 6D). Additionally, significant increases in plasma-free protein S activity levels (*p* = 0.004) were recorded in the second month for the high-dose GTE group (Figure 6E).

#### 3.3.4. Platelet Aggregation

Following induction with an agonist ex vivo, the percentage of PLT aggregation obtained from patients with TDT who consumed either placebos or GTE tablets (50 or 100 mg EGCG equivalent) for two months is illustrated in Figure 7A,B. The findings show that, when an ADP agonist was applied, the PLT aggregation percentages remained unchanged in the placebo group, while there was a notable decrease in the two GTE tablet groups in a manner dependent on time. These reductions were particularly significant during the second month (T_2_) (*p* = 0.003 and 0.001, respectively) when compared with the initial levels (T_0_) (Figure 7A). In a similar vein, when a collagen agonist was utilized, the treatment of patients with TDT with two GTE tablets led to a decrease in PLT aggregation in a time-dependent fashion. Accordingly, significant differences emerged in the second month for the low-dose GTE tablet group (*p* = 0.007), whereas notable differences were found in both months 1 and 2 for the high-dose GTE tablet group (*p* = 0.003) (Figure 7B).

#### 3.3.5. Body Iron and Liver Function Status

The levels of iron status parameters such as PI, TIBC, and TS did not show significant differences between the placebo and the two groups receiving GTE; however, changes were observed after two months of interventions (Figure 8A–C, respectively). It is important to note that the intake of PB and GTE tablets (50 and 100 mg EGCG equivalent each) over a period of two months did not lead to a reduction in their plasma ferritin levels (Figure 8D). Accordingly, oral intakes of low- and high-dose GTE products did not mitigate iron overload status in patients with β-thalassemia.

The liver function parameters, including total protein, albumin, globulin, AST, ALT, and ALP, along with the changes noted in the plasma of TDT participants who were administered either the placebo or GTE tablets over a two-month duration, are detailed. The levels of liver function parameters such as total protein, albumin, globulin, AST, ALT, and ALP, as well as alterations in the plasma of TDT individuals who took the placebo or GTE tablets over the span of two months, are depicted in Table 4. In general, there were no notable differences in any of these parameters across the three groups or at various time intervals. However, almost all parameters remained within healthy limits. This was mainly the case, with the exception being the AST activity levels in these three groups, which tended to be slightly elevated beyond the healthy range.

## 4. Discussion

Normally, PLT dense granules are known to contain ADP, ATP, GTP, serotonin, histamine, and Ca^2+^, for which granule secretion can amplify the cascade of PLT activation, adhesion and aggregation, inflammation, atherosclerosis, angiogenesis, and wound healing. Soluble PLT agonists (e.g., ADP, TXA_2_, and thrombin) function to activate PLT, recruit circulating PLT to vascular injury sites, bind to G-protein-coupled receptors on the PLT membrane, and activate downstream signaling pathways. These events can lead to Ca^2+^ mobilization, stimulation of tyrosine kinase and phospholipase activities, as well as PLT aggregation. Mechanistically, ADP interacts with PLT purinergic P2X and P2Y receptors and stimulates PLT aggregation; inversely, thienopyridines, ticlopidine, and clopidogrel bind to these receptors covalently and irreversibly while inhibiting PLT aggregation [6]. Exposed collagen at vascular injury sites can bind to the glycoprotein I6-IX-V receptor on the PLT membrane using the von Willebrand factor (vWF) as a linker. Both platelet activation and hyperactivity are notably linked to vascular disease, hemostatic processes, and thrombosis.

Pathophysiologically, PLT hyperactivity and hyper-aggregation can occur when levels of stimuli, such as those of thromboxane A2 (TXA_2_), Col, or ADP, are below a certain threshold for full, irreversible PLT aggregation, while increased vWF levels can lead to hypercoagulability in patients with thalassemia, particularly in splenectomized cases [22]. Many confounding factors can be attributed to abnormal hemostasis in patients with thalassemia, including microparticles (MPs) obtained from PLT, leucocytes and vascular endothelial cells, RBC vesicles, plasminogen activator inhibitor 1, RBC phosphatidylserine exposure, plasma NO^•^, β_2_-thromboglobulin (β_2_-TG), and splenectomy status [23,24]. For instance, patients with splenectomized HbE/β-thalassemia exhibited significantly higher levels of plasma β_2_-TG, ADP- and thrombin-induced PLT aggregation, and thrombin–antithrombin III complex levels than healthy persons and patients that were non-splenectomized, while β_2_-TG and thrombin may cause PLT hyperactivity [24]. Apparently, circulatory MPs, activated/procoagulant PLT, and the leucocyte-forming micro-aggregates present in the plasma compartment can contribute to a hypercoagulable state in patients with HbE/β-thalassemia [25]. Iron dextran loading accelerated occlusive thrombosis in mice but did not affect plasma coagulation and ADP-induced PLT aggregation, whereas ingestion of di-cysteine as a free-radical scavenger completely mitigated a thrombotic effect [26]. Accordingly, iron dextran loading increased COXs-catalyzed production of TXB_2_ and Ca^2+^ uptake in PLT in rats, while vitamin E treatment decreased TXB_2_ and PLT aggregation levels [27]. Likewise, PLT aggregation levels were increased by oxidative stress in patients with thalassemia [23].

Patients with thalassemia exhibit a significant prevalence of both venous and arterial thrombosis. Specifically, around 4% of individuals with β-thalassemia major and 29% of those with thalassemia intermediate have reported experiencing thrombotic events [1]. Additionally, the patients who had undergone a splenectomy or who had been inadequately transfused have shown significant increases in the prevalence of thrombotic events [28,29]. This is likely because their abnormal RBCs contained irregular surfaces of phosphatidylserine exposure on the membrane’s outer leaflet, which would then exhibit procoagulant properties [30]. Moreover, their additional common complications may involve the activation of platelets and endothelial cells, which can lead to thromboembolism and hypercoagulation [31]. Moreover, numerous studies have indicated an elevation in coagulant markers within the plasma, including the thrombin–antithrombin complex and the prothrombin fragment (F1 + 2), particularly in patients who were splenectomized and diagnosed with thalassemia [32,33,34]. It is crucial to note that protein C and protein S, both vitamin K-dependent proteins found in the plasma compartment, function together as an anticoagulant system to inhibit excessive blood coagulation, with protein S acting as the primary cofactor for protein C. Nevertheless, reduced levels of these anticoagulant proteins, such as protein C and protein S, have been observed in patients with thalassemia, potentially leading to a hypercoagulable condition [35,36].

Interestingly, green tea offers nutraceutical, biological, and pharmacological properties, indicating their potential cosmetic and health effects [37]. Previous studies have demonstrated that the EGCG found in GTE has iron-chelating and iron-binding capacities [10], anti-hemolytic activity [38], interferes with platelet COXs-catalyzed TXA_2_ formation [13], and inhibits ADP-induced PLT aggregation [39]. The anticoagulant properties of GTE or any other active ingredient isolated from green tea have been the subject of numerous prior investigations, which is particularly true of catechins [13,14]. A previous study demonstrated that an oral intake of green tea EGC at 0.5 and 1.0 g/kg/day demonstrated anticoagulant properties by extending aPTT values and bleeding time while not affecting PT values [14]. Furthermore, Elkhalifa and associates found that healthy individuals who ingested green tea daily for over a year showed increased PT and aPTT values in comparison to the control group [40]. Additionally, other catechins, including ECG and catechin 3-gallate, have been noted for their ability to promote platelet aggregation [41]. On the other hand, it was discovered that EGCG in green tea inhibits PLT adhesion and aggregation without altering PLT surface indicators like P-selectin 1 or activating the glycoprotein IIb/IIIa complex or procaspase activating compounds 1 [15].

A myriad of factors, such as genetic variants, gender, age, sex steroid hormones, plasma free Ca^2+^ concentration, and anti-platelet drugs, can affect PLT reactivity. Of note, PLT reactivity is increased in females, aging status, and ADP depletion, and it is lowered by ASA and P2Y_12_ antagonists [42]. In addition, patients who were splenectomized and non-splenectomized with β-thalassemia HbE had higher PLT aggregation than healthy subjects, and the patients who were splenectomized showed a significantly higher PLT aggregation than the patients who were non-splenectomized, implying a role of splenic absence in PLT hyper-aggregation [43]. In the present work, subjects with β-thalassemia stopped their regular chelation for 72 h before blood collection; therefore, no iron chelators remained in blood circulation and interfered with the study. The number of male PRP and female PRP samples used in the in vitro study and those collected from the clinical study were not significantly different. Without treatment, levels of PLT aggregation in the N-PRP, T-PRP, and T-PRP^ samples were found to be comparable and non-significantly different, whereas the PLT aggregation changes varied between the two groups and among the GTE treatment. Importantly, the GTE treatment exhibited significantly higher inhibitory PLT aggregation than the EGCG treatment, possibly due to the synergistic effects of other active phytochemicals, and the inhibitions were significantly higher in the PRP from male subjects and the PRP from female PRP subjects, possibly due to the sex hormone effect. Apparently, GTE at a dose of 1.25 mg EGCG equivalent inhibited PLT aggregation in healthy PRP induced by ADP and Col but had no effect on thalassemic PRP. Potentially, EGCG, along with some other phenolic compounds in GTE, could chelate plasma Ca^2+^ by contributing to the coagulation cascade with PLT Ca^2+^ acting as a second messenger participating in cellular signal transduction, while redox-active Fe^2+^ in thalassemic plasma could promote ROS generation. In addition, two classical iron chelators, namely 2,2′-dipyridyl and 4,4′-dipyridyl, have been acknowledged for their inhibitory effects on platelet COXs but had no effect on PLT per se [44]. Notably, high amounts of bioactive polyphenolic compounds, particularly EGCG in green tea, are also known to be responsible for these PLT inhibitory effects [41]. Concordantly, consumption of GTE improved PLT aggregation and hypercoagulability in patients with TDT, likely by increasing the anticoagulant protein S and C levels and activities but neither were influenced nor additionally reduced by antiplatelet ASA activity [15,45]. We firstly reported inhibitory effects of GTE on PLT aggregation and blood coagulation in Thai patients with TDT (https://assets-eu.researchsquare.com/files/rs-2583996/v1/6eb06d46-0f97-4063-9f66-ee1fc5094876.pdf?c=1711085943, accessed on 1 April 2024). In the current clinical investigation, we have shown that the peak levels of platelet (PLT) aggregation induced by ADP and collagen (Col) were influenced by the intake of green tea extract (GTE) tablets, containing 50 mg and 100 mg of EGCG equivalent, by approximately 18–22% and 17–25%, respectively. Consequently, these results suggest that EGCG may possess antiplatelet properties by obstructing PLT signaling pathways activated by both proteolytic and non-proteolytic agonists.

Regarding the bioavailability of catechins from green tea, healthy participants who consumed green tea (3 g, equating to 0.9 g of total catechins) showed a rapid increase in blood catechin levels (13%) from their baseline, peaking at 2.3 h before quickly declining, with a half-life elimination rate of 4.8 h [46]. In this context, green tea catechins, such as EGC (the main catechin), GC, EC, C, EGCG, GCG, ECG, and CG, were primarily found in the protein-rich fractions and high-density lipoproteins within the plasma compartment [47,48]. Subsequently, the ingested green tea catechins underwent biotransformation into various metabolites via processes such as methylation, glucuronidation, sulfation, and ring fission in the plasma, with 15 metabolites of EC and EGC primarily eliminated through urine [49]. Our recent comprehensive analysis using UHPLC/EIS QTOF/MS has revealed at least twelve metabolites were detected in the serum of patients with TDT following consumption of GTE tablets for 30 days [16].

Interestingly, GTE tablets resulted in a notable increase in the activities of protein C and protein S by around 28% and 19%, respectively, along with a 12% rise in free protein S levels in the plasma of TDT participants over time. Consequently, it was concluded that the parent EGCG and its metabolites found in the plasma may enhance the functions of protein S and protein C in the plasma environment and/or boost their biosynthesis in liver cells. In comparison, both GTE and EGCG were observed to chelate reactive iron and diminish oxidative stress in liver tissue samples [50]. Nonetheless, we did not observe any significant alterations in liver function parameters in the two intervention groups receiving GTE tablets. It is likely that GTE had no direct effect on liver damage or total protein synthesis activities but rather indirectly increased the hepatocyte capacity of the participants, who had been relieved of iron-overloaded stress, to produce protein C and protein S. At this point, GTE did not have a direct impact on liver damage or the overall synthesis of proteins. Instead, it indirectly enhanced the ability of participants’ hepatocytes to produce protein C and protein S, particularly in those who experienced relief from oxidative stress due to iron overload. Currently, the two phenomena associated with the elevated levels and functions of anticoagulant proteins in patients with thalassemia following the intake of GTE tablets remain inconclusive.

In advantages, GTE potentially mitigated PLT aggregation in both healthy and thalassemic PRP induced by ADP and Col. The product also abrogated PLT activation by chelating redox-active iron, scavenging free radicals, and interfering with PLT cytosolic Ca^2+^ mobilization, consequently inhibiting platelet activation and aggregation in patients with β-thalassemia. The confounding factors of this inhibition can be attributed to the active compounds and individual plasma conditions, as well as the subject’s responsiveness to the compounds and the agonist. Accordingly, frequent consumption of GTE could be beneficial for human health with regard to its preventive effects on platelet aggregation properties, cardiovascular diseases, and pulmonary thromboembolism. In terms of any limitations, other agonists, including TXA2, thrombin, thrombin receptor activator peptide-6, and TXA2 analogue (U46619), should be used to assess anti-platelet activity by GTE in PRP samples. Consequently, larger PRP volumes and sample sizes would be needed to determine the relevant inhibitory effects. In addition, the length of the intervention may have been inadequate, the number of enrolled patients with TDT could have been too small, and it is likely that the participants should have been hospitalized for the entire study period.

## 5. Conclusions

This study’s findings indicate the promising anti-platelet aggregation potential of green tea (possibly with regard to EGCG content) in PRP samples collected from healthy subjects, which were better than those collected from patients with TDT with iron overload. Platelet inhibitory potency was attributed to active phytochemicals, the plasma environment, and any individual responsiveness. Supportive treatment utilizing EGCG-rich green tea extract (GTE) tablets in patients with thalassemia major (TDT) experiencing iron overload may enhance the activities and concentrations of key anticoagulant proteins, specifically protein C and protein S. Furthermore, these GTE tablets demonstrated significant inhibitory effects on platelet aggregation induced by ADP and collagen, in addition to their antioxidant, reactive oxygen species (ROS) scavenging, and iron-chelating capabilities. Notably, the intake of GTE tablets resulted in a substantial reduction in plasma ferritin levels, suggesting an improvement in conditions of iron overload. Collectively, EGCG-rich GTE presents a potential supportive intervention for mitigating or preventing a hypercoagulable state in individuals with thalassemia. Nonetheless, further investigation is necessary to ascertain the extent of the increase in both the levels and functions of protein C and protein S following the consumption of GTE tablets. Further experiments will need to be conducted to explain what confounding factors, mechanisms, and pathways are attributed to the inhibition of PLT aggregation.

## Figures and Tables

**Figure 1 foods-13-03864-f001:**
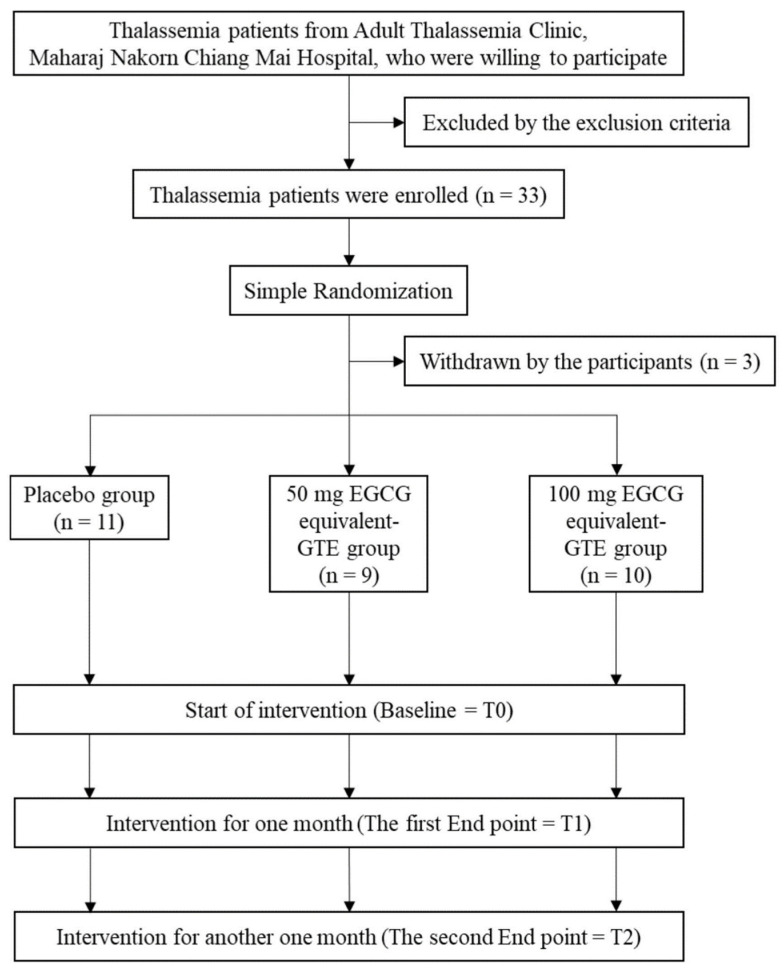
CONSORT flow diagram of the clinical study.

**Figure 2 foods-13-03864-f002:**
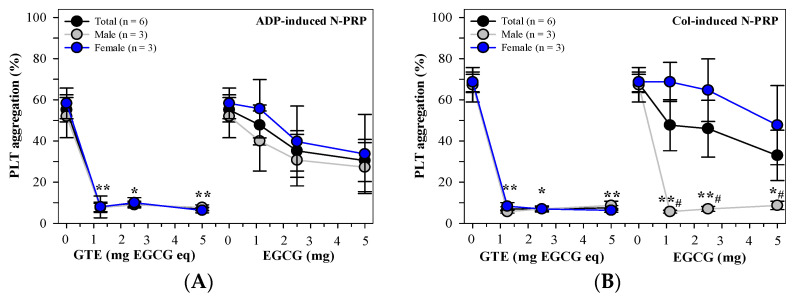
Mean ± SEM values of PLT aggregation in N-PRP samples treated with PBS, GTE (1.25–5 mg EGCG eq) or EGCG (1.25–5 mg), which were induced by (**A**) 5 μM of ADP or (**B**) 2 μg/mL of Col agonist. Accordingly, * *p* < 0.05, ** *p* < 0.01 when compared without treatment; ^#^ *p* < 0.05 when compared between genders.

**Figure 3 foods-13-03864-f003:**
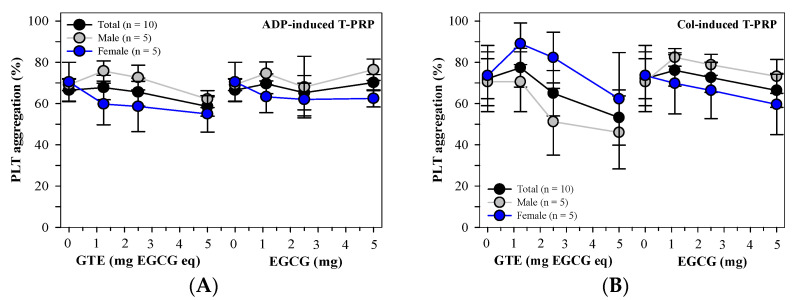
Mean ± SEM values of PLT aggregation in T-PRP samples treated with PBS, GTE (1.25–5 mg EGCG eq), and standard EGCG (1.25–5 mg) that had been induced by (**A**) 5 μM of ADP or (**B**) 2 μg/mL of Col agonist.

**Figure 4 foods-13-03864-f004:**
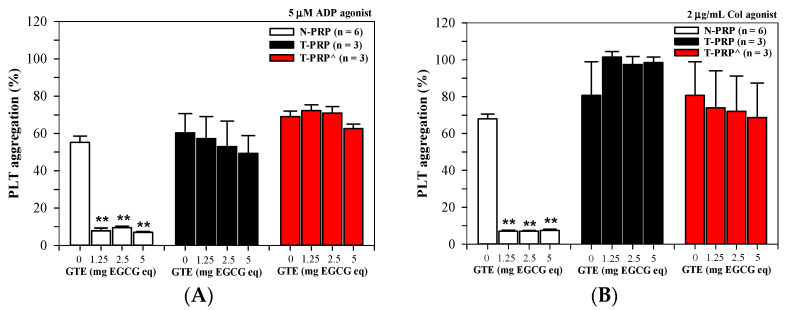
PLT aggregation in N-PRP, T-PRP, and T-PRP^ samples treated with GTE (1.25–5.0 mg EGCG eq) that were induced by (**A**) 5 μM of ADP and (**B**) 2 μg/mL of Col. Data are expressed as mean ± SEM values, which ** *p* < 0.01 when compared without treatment.

**Figure 5 foods-13-03864-f005:**
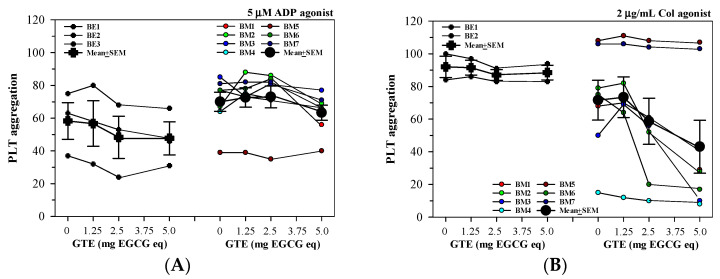
PLT aggregation in T-PRP samples obtained from patients with BE, BM, and H thalassemia treated with PBS, GTE (1.25–5.0 mg EGCG eq), and standard EGCG (1.25–5.0 mg) that were induced by (**A**) 5 μM of ADP and (**B**) 2 μg/mL of Col. Data are expressed as individual and mean ± SEM values.

**Figure 6 foods-13-03864-f006:**
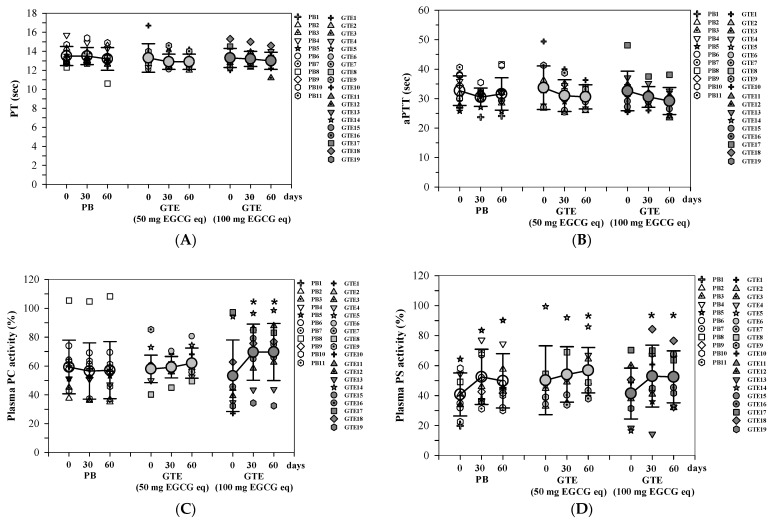
PT (**A**), aPTT (**B**), PC (**C**), PS (**D**), and free PS activity (**E**) values in plasma of patients with TDT who had consumed a PB tablet (n = 11), a single GTE (50 mg EGCG eq) tablet (n = 9), and double-GTE (50 mg EGCG equivalent) tablets (n = 10) tablet once daily for 60 days. Data are expressed as individual and mean ± SD values. Accordingly, * *p* < 0.05 when compared with day 0.

**Figure 7 foods-13-03864-f007:**
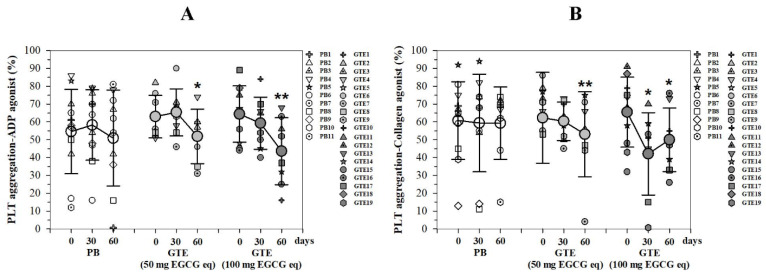
Percentages for PLT aggregation in PRP samples of patients with TDT who had consumed PB tablet (n = 11), a single GTE (50 mg EGCG eq) tablet (n = 9), and double-GTE tablets (50 mg EGCG eq) tablet (n = 10) once daily for 60 days after induction by the addition of an agonist, either 5 μM of ADP (**A**) or 2 mg/mL of collagen (**B**) in vitro. Data are expressed as individual and mean ± SD values. Accordingly, * *p* < 0.05 and ** *p* < 0.01 when compared with T0.

**Figure 8 foods-13-03864-f008:**
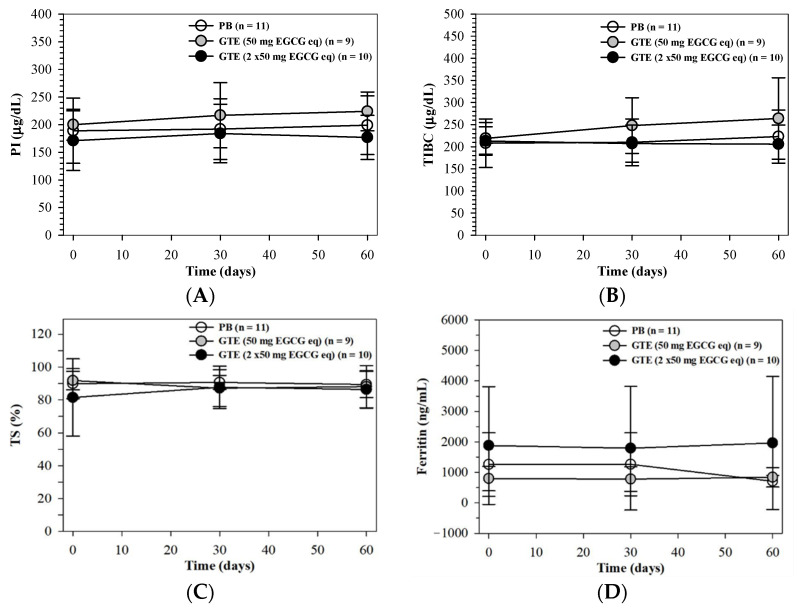
Levels of PI (**A**), TIBC (**B**), TS (**C**), and Ft (**D**) in the plasma of patients with TDT who had consumed a PB tablet (n = 11), a single GTE tablet (50 mg EGCG equivalent) (n = 9), and double-GTE tablets (100 mg EGCG eq) (n = 10) once daily for 60 days. Data are expressed as mean ± SD values.

**Table 1 foods-13-03864-t001:** Demographic characteristics and some blood parameter levels of subjects who were healthy and subjects with TDT at the time of blood collection. Data are expressed as absolute numbers, percentage, ranges, or mean ± SD values.

Information	Healthy Subjects	Patients with Thalassemia
Gender:	6 (3 male, 3 female)	10 (5 male, 5 female)
Age (years):	26.8 ± 1.0 (25–28)	32.6 ± 7.2 (20–46)
BW (kg):	63.0 ± 8.9	52.6 ± 7.0
BMI (kg/m^2^):	33.1 ± 7.2	32.9 ± 3.3
Type of thalassemia: n (%)		
BM	0	7 (70.0)
BE	0	3 (30.0)
Iron chelator: n (%)		
DFP	0	7 (70.0)
DFX	0	1 (10.0)
DFP + DFO	0	1 (10.0)
DFO + DFX	0	1 (10.0)
ASA: n (%)		
Taking	0	7 (70.0)
Not taking	6 (100.0)	3 (30.0)
Splenectomy: n (%)		
Performed	0	7 (70.0)
Not performed	6 (100.0)	3 (30.0)
Liver span (cm):	0	12.1 ± 1.8
PLT numbers (×10^5^ cells/mm^3^)	2.36 ± 0.28	5.36 ± 2.74
Ft (ng/mL)	99 ± 9	1098 ± 484

**Table 2 foods-13-03864-t002:** Demographic information of patients with TDT participating in this study, who had consumed placebo (PB1-PB11), 50 mg EGCG equivalent-GTE (GTE1-GTE9), or 100 mg EGCG equivalent-GTE (GTE10-GTE19) tablets. Data are expressed as absolute or mean ± SD values.

Code	Sex	Type	Age (y)	IC	ASA	SPX	BW (kg)	Ht (cm)	BMI (kg/m^2^)	Liver Span (cm)
PB1	F	BE	40	DFP	Y	Y	46	145	31.7	10
PB2	M	BE	38	DFP	Y	Y	47	168	28.0	10
PB3	M	BM	31	DFP	Y	Y	42	158	26.6	13
PB4	F	BM	31	DFX	Y	Y	34	140	24.3	12
PB5	M	BM	30	DFP	Y	Y	43	160	26.9	12
PB6	M	BM	25	DFP	Y	Y	54	160	33.8	16
PB7	F	BE	47	DFP	N	N	47	160	29.4	12
PB8	F	AEBart	54	DFP	N	N	46	148	31.1	10
PB9	M	BE	54	DFP	N	N	53	160	33.1	15
PB10	M	BE	39	DFP	N	N	58	170	34.1	13
PB11	F	BE	49	DFP	N	N	47	156	30.1	17
	6M, 5F	4BM, 6BE, 1AEBart	39.8 ± 10.1	10DFP, 1DFX	6/11	6/11	47.0 ± 6.5	157 ± 9	29.9 ± 3.2	12.7 ± 2.4
GTE1	F	AE Bart	62	DFP	Y	Y	30	135	22.2	10
GTE2	M	BM	35	DFP + DFO	Y	Y	44	150	29.3	12
GTE3	F	BE	27	DFP	Y	Y	48	158	30.4	13
GTE4	F	BM	28	DFP	Y	Y	38	145	26.2	10
GTE5	F	BM	22	DFO + DFX	Y	Y	45	145	31.0	10
GTE6	M	BE	22	DFP	Y	Y	55	161	34.2	14
GTE7	F	BE	39	DFP	Y	Y	39	152	25.7	10
GTE8	F	BM	24	DFP	Y	Y	40	156	25.6	10
GTE9	M	BM	34	DFP	N	N	42	151	27.8	10
	3M, 6F	5BM, 3BE, 1AEBart	32.6 ± 12.6	7DFP, 1DFO + DFP, 1DFO + DFX	8/9	8/9	42.3 ± 7.0	150 ± 8	28.1 ± 3.6	11.0 ± 1.6
GTE10	F	BM	36	DFO + DFX	Y	Y	48	149	32.2	12
GTE11	F	BE	33	DFP	Y	Y	46	150	30.7	13
GTE12	F	BM	27	DFP	Y	Y	48	150	32.0	10
GTE13	M	BM	25	DFO + DFX	Y	Y	70	175	40.0	13
GTE14	M	BE	30	DFP	Y	Y	55	153	35.9	10
GTE15	M	BE	29	DFP	Y	Y	54	175	30.9	10
GTE16	M	BE	47	DFP	Y	Y	58	175	33.1	18
GTE17	F	BE	21	DFP	N	N	55	156	35.3	10
GTE18	F	BM	37	DFO + DFP	N	N	60	165	36.4	13
GTE19	M	BE	30	DFP	N	N	50	166	30.1	10
	5M, 5F	4BM, 6BE	31.5 ± 7.3	7DFP, 2DFO + DFX, 1DFO + DFP	7/10	7/10	54.4 ± 7.2	161 ± 11	33.7 ± 3.2	11.9 ± 2.6

**Table 3 foods-13-03864-t003:** PLT index values in patients with TDT who enrolled in this study, who had consumed placebos, 50 mg EGCG equivalent-GTE or 100 mg EGCG equivalent-GTE tablets. Data are expressed as absolute or mean ± SD values.

PLT Indices	PB (n = 11)			GTE (50 mg EGCG eq) (n = 9)			GTE (100 mg EGCG eq) (n = 10)		
	T0	T1	T2	T0	T1	T2	T0	T1	T2
Number (×10^5^/mm^3^)	5.97 ± 0.66	5.92 ± 0.68	5.90 ± 0.94	5.97 ± 1.08	6.05 ± 1.37	6.07 ± 1.49	5.77 ± 0.89	5.11 ± 1.21	4.78 ± 1.44
MPV (fL)	10.00 ± 0.42	10.26 ± 1.95	10.30 ± 0.40	10.00 ± 5.72	9.95 ± 0.37	9.90 ± 0.42	9.95 ± 0.56	9.86 ± 0.52	9.64 ± 0.43
PDW (%)	10.88 ± 1.42	10.80 ± 1.44	10.93 ± 1.67	11.00 ± 2.85	10.94 ± 2.46	11.10 ± 1.73	11.20 ± 1.40	10.60 ± 1.08	10.85 ± 1.17
Pct (%)	0.69 ± 0.10	0.63 ± 0.09	0.60 ± 0.06	0.67 ± 0.13	0.62 ± 0.12	0.64 ± 0.19	0.66 ± 0.14	0.62 ± 0.14	0.64 ± 0.10
IPF (%)	1.83 ± 0.86	1.86 ± 0.69	1.74 ± 0.93	1.80 ± 0.85	1.81 ± 0.61	1.88 ± 0.82	1.70 ± 0.71	1.58 ± 0.48	1.60 ± 0.32

**Table 4 foods-13-03864-t004:** Levels of TP, Alb, Glo, ALP, AST, and ALT in the plasma of patients with TDT who had consumed a PB tablet (n = 11), a single GTE tablet (50 mg EGCG equivalent) (n = 9), and double-GTE tablets (100 mg EGCG eq) (n = 10) once daily for 60 days. Data are expressed as mean ± SD values.

Parameter	PB (n = 11)			GTE (50 mg EGCG eq) (n = 9)			GTE (100 mg EGCG eq) (n = 10)		
	T0	T1	T2	T0	T1	T2	T0	T1	T2
TP (g/dL)	8.1 ± 0.6	8.1 ± 0.7	8.0 ± 0.6	8.6 ± 0.6	8.8 ± 0.7	8.5 ± 0.9	8.5 ± 0.6	8.4 ± 0.6	8.7 ± 0.7
Alb (g/dL)	4.4 ± 0.3	4.3 ± 0.3	4.3 ± 0.4	4.6 ± 0.2	4.5 ± 0.6	4.3 ± 0.7	4.4 ± 0.4	4.4 ± 0.1	4.6 ± 0.2
Glo (g/dL)	3.70.6	3.80.6	3.60.6	4.10.7	4.31.1	4.30.9	4.10.6	4.00.7	4.00.7
AST (U/L)	87 ± 34	83 ± 25	86 ± 28	105 ± 42	97 ± 37	116 ± 56	89 ± 24	87 ± 24	96 ± 32
ALT (U/L)	27 ± 9	26 ± 11	11 ± 3	46 ± 28	46 ± 33	58 ± 48	28 ± 12	32 ± 15	32 ± 14
ALP (U/L)	20 ± 13	15 ± 6	8 ± 2	39 ± 21	37 ± 27	48 ± 46	24 ± 11	31 ± 20	30 ± 13

## Data Availability

The original contributions presented in the study are included in the article/Appendix A, further inquiries can be directed to the corresponding author. This manuscript has not undergone peer review, nor has it been considered elsewhere prior to publication. A preprint of this article has been posted from Research Square (PPR: PPR634002) on 23 March 2023 with the following DOI link: https://doi.org/10.21203/rs.3.rs-2583996/v1 (accessed on 1 April 2024).

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
