# Peer review of "Green Tea Epigallocatechin 3-Gallate Reduced Platelet Aggregation and Improved Anticoagulant Proteins in Patients with Transfusion-Dependent β-Thalassemia: A Randomized Placebo-Controlled Clinical Trial"

_foods, 2024, doi:10.3390/foods13233864_

Round 1

Reviewer 1 Report

Comments and Suggestions for Authors

There are minor errors in the article

1. Line 69-72 “Intriguingly, green tea extract (GTE) contains predominant polyphenolics including catechin (C), epigallocatechin (EGC), epicatechin-3-gallate (ECG), epicatechin (EC), epigallocatechin-3-gallate (EGCG), CGA, caffeic acid and caffeine (CF), of which EGCG is the most abundant (59% of total catechins) [9]”.   Should add    chlorogenic acid (CGA)

 2. Fig 4  Should to   add 5 µM ADP on top  of Fig А

Major comments: 

1. Lines 296-298 “Accordingly, the GTE and EGCG treatments did not influence PLT aggregation in T-PRP (n = 10) induced by 5 µM ADP and 2 µg/mL Col agonists when compared with those that had not undergone treatment”. This sentence contradicts the previous one and results Fig3 D,E,F.

2.Lines 308-310 “As has been shown in Figures 4A, GTE (1.25-5.0 mg EGCG equivalent) treatments inhibited PLT aggregation slightly in T-PRP and hardly in the PRP of splenectomized thalassemia (T-PRP^) patients that had been induced by 5µM ADP, whereas PLT aggregation was inhibited more markedly in those induced by 2 µg/mL Col, while the treatments inhibited PLT aggregation in N-PRP considerably (Figure 4B)”. This description of the results is not consistent with the data presented on Fig 4A,B

3. In the results ( 3.2.2 ; 3.2.3) effects of extract and EGCG on   the aggregation parameter is described by the expressions ‘increases hardly’ or ‘decreases considerably’ or “less effective,  more effective” but no numerical values are given.  Only in the discussion, in the  describtion of  PLT aggregation and levels and activity of protein C  , numerical data are given but without statistical analysis.

4. In Figure 6 D,E - It is not clear where the activity of protein S and  where its С  level are represented.  Does  figure 6C  represent the activity or the level of protein C ? The text says level and the figure shows  activity.  

5. On the graph 8 parameters  (PI and TIBC) change in the opposite direction depending on the dose : at 50 µg increased and at 100 µg decreased. Why? What is explanation ? 

6.The authors showed that EGCG acts more strongly than extract at equivalent EGCG concentrations and in the N-PRP and T-PRP models. The authors also showed a difference in the effect of extract and EGCG when using PRP from male and female, but did not specifically discuss the reasons or possible mechanisms behind these effects.

Reviewer 2 Report

Comments and Suggestions for Authors

1. The sample size of participants included was small and the sample was not geographically representative and representative of the population.

2. Precise data should be used to demonstrate the strength of the study results. For example, do not simply state that "plasma ferritin levels were reduced", but emphasize how these changes may affect patient outcomes.

3. How potential confounding factors (such as differences in iron chelation therapy) were handled was not fully explained.

4. The legends of some graphs (such as platelet aggregation data) were unclear.

Comments on the Quality of English Language

The writing is not good enough, the writing is not smooth, and the wording and sentence structure are not accurate.

Reviewer 3 Report

Comments and Suggestions for Authors

The manuscript titled “Green tea epigallocatechin 3-gallate reduced platelet aggregation and improved anticoagulant proteins in transfusion-dependent β-thalassemia patients: a randomized placebo-controlled clinical trial” presented by Touchwin Petiwathayakorn  discussed the Green tea bioactive epigallocatechin 3-gallate effect on platelet aggregation and its role as anticoagulant proteins. Overall manuscript written well but I feel the manuscript has some flows that’s needs to rectified in further submission.  A detailed reviewer report mark in pdf file, kindly refer to pdf page.

General comment:

·         At several places there is spacing issue, authors are suggested to correct it

·         Minor English and grammar mistakes are present that needs to be thoroughly checked to improve the quality of the manuscript.

Comments on the Quality of English Language

Minor editing of English language required.

Round 2

Reviewer 1 Report

Comments and Suggestions for Authors

I found no changes in the new version of the manuscript in response to my main comments

For example,” Response 1: We apologize for the mistakes. We have now corrected them (Pages 9-10, Lines 344-347, the Results section). New sentences are “Accordingly, the GTE and EGCG treatments did not influence PLT aggregation in T-PRP (n = 10) induced by 5 M ADP (Figure 3C) and 2 g/mL Col (Figure 3F) agonists when compared with those that had not undergone treatment.”

 In the new version  lines 344-347  are on  page 11  and  sentence is “Additionally, a low-dose ASA regimen was 344 provided to a portion of the participants who had undergone splenectomy (70.0%) to mit-345 igate the risk of thrombosis. The average age, liver span, body weight, and body mass 346 index were determined to be statistically insignificant and fell within the normal range for 347 all subjects.”

This applies to other comments (2,4.5.6)

Author Response

Yours sincerely,

Professor Somdet Srichairatanakool, PhD.

Reviewer 2 Report

Comments and Suggestions for Authors

The current version of the manuscript can be accepted now.

Comments on the Quality of English Language

Minor editing of English language required.

Author Response

(The authors gave the same response as above.)
